# Light-driven flow synthesis of acetic acid from methane with chemical looping

Wenqing Zhang[1,2], Dawei Xi [1], Yihong Chen[1], Aobo Chen[1], Yawen Jiang[1], Hengjie Liu [1], Zeyu Zhou[3,4], Hui Zhang [3,4], Zhi Liu [3,4], Ran Long [1] ✉ & Yujie Xiong [1,2,5] ✉

Oxidative carbonylation of methane is an appealing approach to the synthesis of acetic acid but is limited by the demand for additional reagents. Here, we report a direct synthesis of $CH_3COOH$ solely from $CH_4$ via photochemical conversion without additional reagents. This is made possible through the construction of the $PdO/Pd-WO_3$ heterointerface nanocomposite containing active sites for $CH_4$ activation and C–C coupling. In situ characterizations reveal that $CH_4$ is dissociated into methyl groups on Pd sites while oxygen from PdO is the responsible for carbonyl formation. The cascade reaction between the methyl and carbonyl groups generates an acetyl precursor which is subsequently converted to $CH_3COOH$. Remarkably, a production rate of 1.5 mmol $g_{Pd}^{-1}$ $h^{-1}$ and selectivity of 91.6% toward $CH_3COOH$ is achieved in a photochemical flow reactor. This work provides insights into intermediate control via material design, and opens an avenue to conversion of $CH_4$ to oxygenates.

Methane conversion into value-added chemicals under mild condition is a promising strategy for maximizing $CH_4$ utilization and mitigating the greenhouse effect[1–5]. In particular, the partial oxidation of $CH_4$ at low temperature (<200 °C) is an attractive approach to generate valuable oxygenates (e.g., $CH_3OH$, HCHO, HCOOH and $CH_3COOH$) while reducing energy input and carbon emission in traditional gas-phase $CH_4$ conversion[6–9]. Among the products, acetic acid ($CH_3COOH$) is an important feedstock of chemical industries. Generally, the synthesis of $CH_3COOH$ from $CH_4$ requires a three-step process involving the production of syngas and methanol, which suffers from extra resource consumption and safety issues[10]. It is thus imperative to develop a green synthetic approach that can directly convert $CH_4$ to $CH_3COOH$. Although recent reports have demonstrated the oxidative carbonylation of $CH_4$ to $CH_3COOH$ in thermocatalytic processes, the

requirement for additional oxidants (e.g., $O_2$ and $H_2SO_4$) and/or CO limits their further applications[11,12]. Moreover, multiple side reactions take place in various reactants to generate undesired products such as HCOOH and $CO_2$, and further limit the selectivity to $CH_3COOH$[13].

Intuitively, photocatalysis should be a potential approach to the green transformation of $CH_4$, in which the ·OH radical derived from water oxidation is the ideal substitute for additional oxidant[14,15]. In fact, the metal-decorated semiconductor photocatalysts, which offer synergistic effects between metal and semiconductor on electronic structure, charge separation and intermediate adsorption, have been demonstrated to be effective for $CH_4$ activation[16–18]. For instance, Pd-based photocatalysts have been reported for the conversion of $CH_4$ into $C_1$ oxygenate products ($CH_3OH$, $CH_3OOH$, HCHO, etc.)[19]. The ·OH radical produced from photocatalytic water oxidation enables the

[1]Hefei National Research Center for Physical Sciences at the Microscale, Collaborative Innovative Center of Chemistry for Energy Materials (iChEM), Key Laboratory of Precision and Intelligent Chemistry, School of Chemistry and Materials Science, National Synchrotron Radiation Laboratory, School of Nuclear Science and Technology, University of Science and Technology of China, Hefei, Anhui 230026, China. [2]Institute of Energy, Hefei Comprehensive National Science Center, 350 Shushanhu Rd, Hefei, Anhui 230031, China. [3]School of Physical Science and Technology, ShanghaiTech University, Shanghai 201203, China. [4]State Key Laboratory of Functional Materials for Informatics, Shanghai Institute of Microsystem and Information Technology, Chinese Academy of Sciences, Shanghai 200050, China. [5]Anhui Engineering Research Center of Carbon Neutrality, College of Chemistry and Materials Science, Key Laboratory of Functional Molecular Solids, Ministry of Education, Anhui Normal University, Wuhu, Anhui 241002, China. ✉e-mail: longran@mail.ustc.edu.cn; yjxiong@ustc.edu.cn

activation of CH$_4$ to generate methyl intermediates (*CH$_3$), which can be stabilized on Pd sites for further reactions[20,21]. However, it still remains a grand challenge to achieve the photocatalytic production of CH$_3$COOH, mainly due to the difficulties of forming carbonyl intermediates and controlling methyl−carbonyl coupling in photocatalysis. The formation of carbonyl intermediates is the key to the production of CH$_3$COOH using CH$_4$ as the sole carbon source, which raises very high requirements for rational construction of catalytically active sites on photocatalysts. Once carbonyl intermediates can be formed from CH$_4$, the carbonylation of CH$_4$ to CH$_3$COOH would no longer need the addition of CO reagent. As demonstrated in thermocatalysis[22], the carbonyl group generated in situ from CH$_4$ oxidation can be coupled with the adsorbed *CH$_3$, leading to CH$_3$COOH formation.

Here, we report that CH$_4$ as the sole carbon source can be directly converted to CH$_3$COOH without additional reagents, by rationally integrating the catalytically active sites for CH$_4$ activation and C−C coupling on material surface. The key is the construction of Pd/PdO heterostructure on WO$_3$ support. The photogenerated holes in WO$_3$ enable oxidation of H$_2$O to ·OH radicals for CH$_4$ activation while Pd sites stabilize *CH$_3$ for further conversion. More importantly, PdO—the active species for CH$_4$ oxidation through the Mar−van Krevelen mechanism[23]—is regarded as the key component for the transformation of CH$_4$ to carbonyl intermediate (*CO) under light irradiation. As such, the carbonylation of CH$_4$ can be achieved through the coupling of methyl and carbonyl intermediates, forming acetyl (CH$_3$CO*) precursor toward the final product of CH$_3$COOH. To facilitate the continuous reaction between methyl and carbonyl intermediates, we design a photochemical flow reaction device with arc-shaped flow channel, in which the flux of *CH$_3$ can react with *CO intermediate continually by fully utilizing PdO and *CH$_3$, to perform the cascade reaction. As a result, the PdO/Pd−WO$_3$ heterointerface nanocomposite with optimal PdO content enables the remarkable selectivity of 91.6%

and production rate of 1.5 mmol g$_{Pd}^{-1}$ h$^{-1}$ toward CH$_3$COOH, providing a feasible strategy for scale-up CH$_4$ conversion.

## Results

### Structural characterization of nanocomposites

In the preparation of nanocomposites, Pd nanoparticles (NPs) are loaded on WO$_3$ nanosheets (Pd/WO$_3$), followed by the further thermal annealing process to decorate PdO species on Pd NPs. The obtained samples are denoted as PdO/Pd−WO$_3$-$x$ where $x$ = 1−5 by increasing the annealing temperature (refer to Methods). The Pd contents are kept constant in these samples, which are confirmed by inductively coupled plasma optical emission spectrometry (ICP-OES) (Supplementary Table 1), to exclude the effect of Pd content on CH$_4$ conversion performance. Transmission electron microscopy (TEM) images reveal that the prepared WO$_3$ nanosheets have the edge lengths of ~170 nm (Supplementary Fig. 1), and the nanoparticles in all samples are highly dispersed on WO$_3$ substrate (Supplementary Fig. 2). The sizes of Pd NPs increase from Pd/WO$_3$ to PdO/Pd−WO$_3$-5 with the annealing temperature raised (Supplementary Fig. 3), implying the incorporation of oxygen atoms into the nanoparticles along with their lattice expansion. The samples are further characterized by X-ray diffraction (XRD) as shown in Supplementary Fig. 4. The diffraction peaks of Pd and PdO are absent in the XRD patterns, indicating that the nanoparticles are highly dispersed at a low loading amount.

To look into the detailed structures, the nanoparticles on WO$_3$ supports are examined by high-resolution TEM (HRTEM). The Pd NPs are decorated with PdO with different oxidation degree by controlling the annealing temperature. As shown in Fig. 1a, the pristine Pd nanoparticle only displays the interplanar distance of 2.2 Å, corresponding to the spacing of Pd (111) planes[24−26]. After the annealing process, the new lattice fringes with a spacing of 2.65 Å appear in the nanoparticles (Fig. 1b and Supplementary Fig. 5), which can be assigned to the (101)

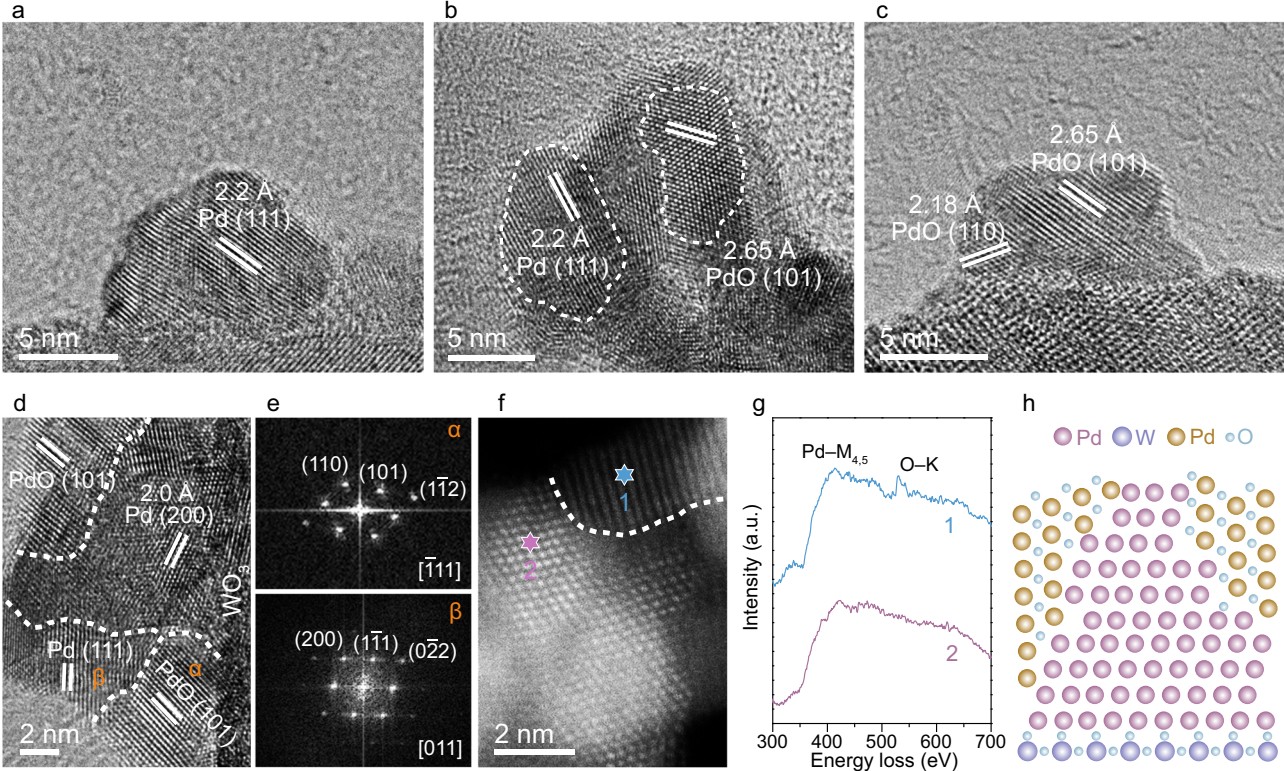

**Fig. 1 | Structural characterization of nanocomposites. a−c** HRTEM images of Pd/WO$_3$ (**a**), PdO/Pd−WO$_3$-2 (**b**) and PdO/Pd−WO$_3$-5 (**c**). **d** Typical HRTEM image of PdO/Pd−WO$_3$-2 sample showing Pd/PdO heterostructure. **e** Corresponding FFT patterns of α and β regions in **d**. **f** HAADF-STEM image of PdO/Pd−WO$_3$-2 sample. **g** EELS spectra collected in the regions 1 and 2 marked in **f**. **h** Structural illustration of PdO−Pd−WO$_3$ heterointerface.

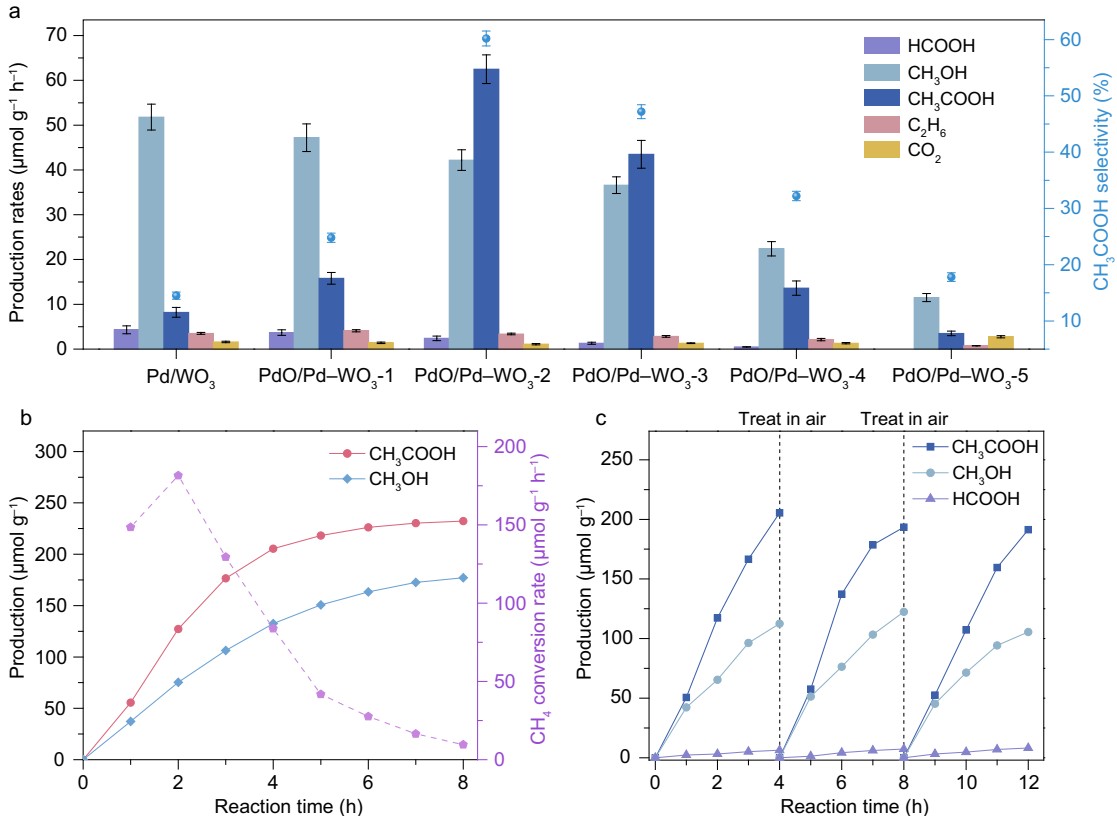

**Fig. 2 | Light-driven CH$_4$ conversion performance of nanocomposites.**
**a** Production rates for light-driven CH$_4$ conversion over Pd/WO$_3$ and PdO/Pd–WO$_3$-1 to PdO/Pd–WO$_3$-5 samples. **b** Time-dependent rates of CH$_3$COOH and CH$_3$OH production as well as CH$_4$ conversion over PdO/Pd–WO$_3$-2 nanocomposite. **c** Reaction-regeneration cycles in CH$_4$ conversion. The error bars represent the standard deviation of the experiments.

planes of PdO[27,28]. Meanwhile, the Pd (111) planes are still observed in the nanoparticles, indicating the existence of Pd/PdO heterostructure in PdO/Pd–WO$_3$-1 to PdO/Pd–WO$_3$-4. As the annealing temperature reaches 450 °C, the Pd NPs are completely transformed to PdO NPs (Fig. 1c). Moreover, the compositions of Pd and PdO species are investigated by X-ray photoelectron spectroscopy (XPS). As shown in Supplementary Fig. 6, the content of Pd$^{2+}$ increases by elevating the annealing temperature, in agreement with the findings from HRTEM images.

The Pd–PdO interface is further resolved meticulously to illustrate the active structure for CH$_4$ conversion. Taking PdO/Pd–WO$_3$-2 as an example, abundant Pd/PdO grain boundaries are observed by the distinguishable lattice parameters of Pd and PdO (Fig. 1d and Supplementary Fig. 7). Figure 1e shows the fast Fourier transform (FFT) diffraction patterns obtained from the α and β regions in Fig. 1d. The FFT pattern with the labels of (110), (101) and (1$\bar{1}$2) in α region fits the tetragonal structure of PdO along the zone axis of [$\bar{1}$11][29]. Meanwhile, in β region, we can also obtain the FFT pattern of Pd along the zone axis of [011] direction belonging to the face-centered cubic (*fcc*) structure with (200), (1$\bar{1}$1) and (0$\bar{2}$2)[30]. Furthermore, the Pd/PdO grain boundary is examined by atomic-resolution high-angle annular dark-field scanning TEM (HAADF-STEM), with their compositions further analyzed via electron energy-loss spectroscopy (EELS). As shown in Fig. 1f and g, in addition to the detected signals of Pd-M$_{4,5}$ edges at both sites 1 and 2, the peak of O-K edge is also recognized at site 1, corresponding to the composition of PdO[31,32]. Taken together, the aforementioned results demonstrate the existence of Pd/PdO heterostructure on WO$_3$ support. During the annealing process, the PdO component appears on Pd NPs under the cooperation of oxygen and support, as nano-islands rather than core-shell structure[33], forming the PdO–Pd–WO$_3$ triple interface (Fig. 1h).

## Performance of light-driven CH$_4$ conversion

Upon acquiring the fine structures, we are now in a position to investigate the efficacy of the PdO/Pd–WO$_3$ nanocomposites in light-driven CH$_4$ conversion. The photochemical measurements are conducted in a quartz reactor under xenon arc lamp irradiation. Pure WO$_3$ nanosheets show sluggish properties for CH$_4$ conversion (Supplementary Fig. 8). After Pd NPs deposition, the CH$_4$ conversion activity over Pd/WO$_3$ is enhanced with CH$_3$OH as the primary product (Fig. 2a). Interestingly, the photochemical performance is significantly altered after the incorporation of PdO species into the Pd/WO$_3$ structure, which exhibits a volcano-like relationship with the amount of PdO. Specifically, the addition of PdO to the samples dramatically boosts the production of CH$_3$COOH. Among the samples, the PdO/Pd–WO$_3$-2 achieves the highest production rate and selectivity toward CH$_3$COOH at 62.5 μmol g$^{-1}$ h$^{-1}$ and 60.2%, respectively. The outstanding performance of converting CH$_4$ to CH$_3$COOH indicates that the Pd/PdO heterostructure enables an efficient C–C coupling process. However, the excessive PdO in nanocomposites hinders the Schottky contact between Pd and WO$_3$, which further reduces photo-induced charge separation efficiency and substantially suppresses photochemical performance (Supplementary Figs. 9 and 10).

To confirm the carbon source of the liquid products, the origin of CH$_3$OH and CH$_3$COOH, as the main products, are traced with $^{13}$C nuclear magnetic resonance ($^{13}$C NMR) spectroscopy by using $^{13}$CH$_4$ as the reactant. As shown in Supplementary Fig. 11, the peaks at 20.5 and 176.7 ppm are attributed to $^{13}$CH$_3^{13}$COOH while the peak at 48.9 ppm is assigned to $^{13}$CH$_3$OH. In addition, the control experiments indicate that no product can be detected in the absence of nanocomposite, light irradiation or CH$_4$ reactant (Supplementary Fig. 12). These results provide the evidence that the primary products indeed originate from light-driven CH$_4$ conversion.

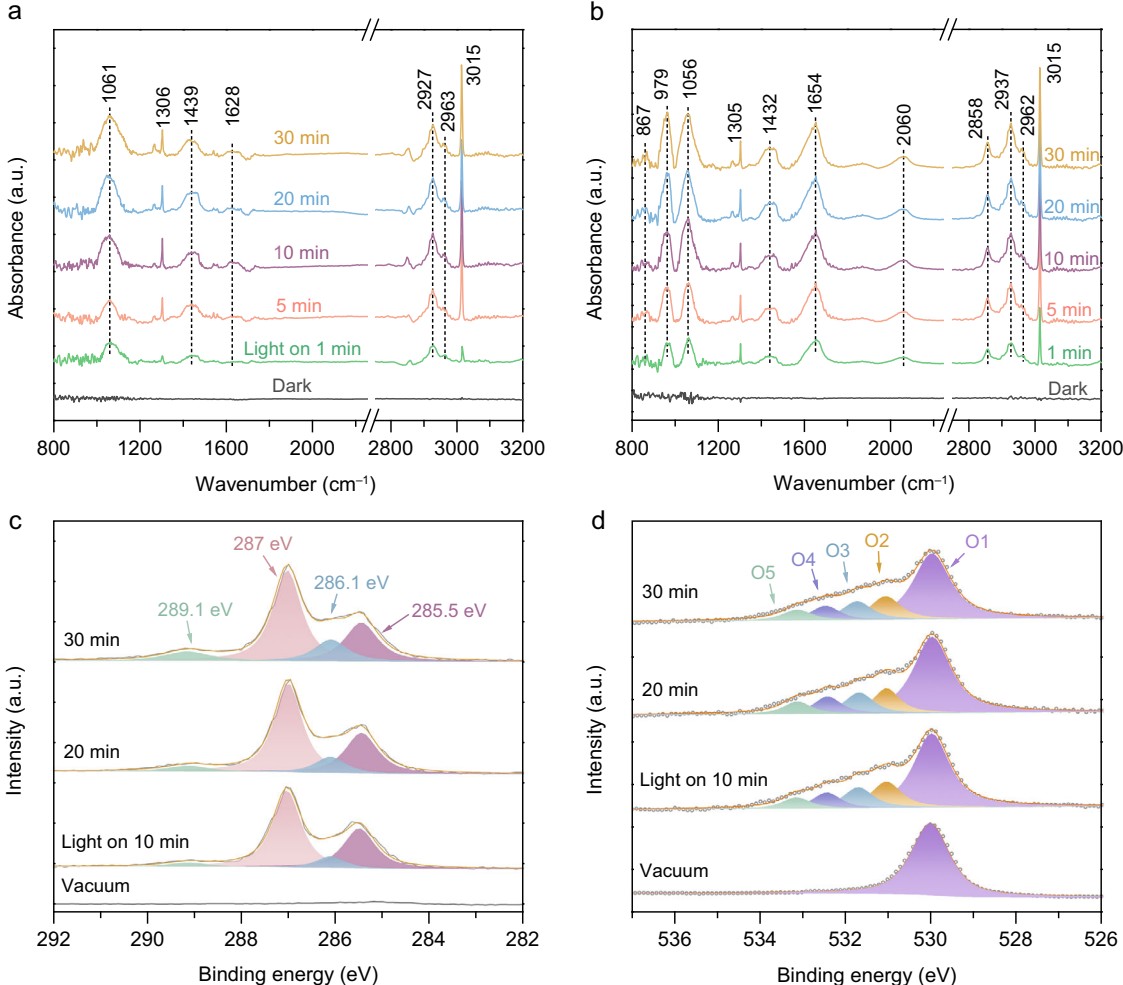

**Fig. 3 | In situ characterizations of the photochemical CH₄ conversion process. a, b** In situ DRIFT spectra for light-driven CH₄ conversion over Pd/WO₃ (**a**) and PdO/ Pd−WO₃-2 (**b**). **c, d** In situ NAP-XPS results of high-resolution C 1 *s* (**c**) and O 1 *s* (**d**) spectra over PdO/Pd−WO₃-2 nanocomposite.

It is worth pointing out that differing from the conventional CH₄ photooxidation requiring extra oxidant addition (e.g., O₂), our reaction system utilizes the reactants of CH₄ and H₂O, in which the ·OH radical produced from water oxidation is the ideal oxidizer for CH₄ activation (Supplementary Figs. 13 and 14)[34]. As displayed in Supplementary Fig. 15, increasing the concentration of O₂ will not promote the generation of liquid products but lead to CO₂ production during the CH₄ photooxidation over PdO/Pd−WO₃-2. The activation of CH₄ by ·OH radicals can produce ·CH₃ as detected by in situ electron paramagnetic resonance (EPR) measurement (Supplementary Fig. 16).

To better evaluate the efficiency, we conduct time-dependent measurement over PdO/Pd−WO₃-2. As shown in Fig. 2b, the production of CH₃COOH and CH₃OH gradually increases in the first 3 h, achieving an impressive CH₄ conversion rate of 181.5 μmol g⁻¹ h⁻¹. However, the performance shows distinct decay when the reaction time exceeds 3 h, which should be ascribed to the consumption of PdO species after the photochemical process (Supplementary Figs. 17 and 18). The constructed Pd/PdO heterostructure is gradually destroyed along with the reaction, which further reduces the efficiency of C−C coupling toward CH₃COOH production. In the meantime, negligible H₂ detection during the reaction suggests that the lattice oxygen of WO₃ is consumed for H₂O production, which also leads to performance decay (Supplementary Figs. 19 and 20). To overcome this limitation, we carry out the regeneration process by heating the nanocomposites in air, in which the consumed oxygen (i.e., PdO on Pd and lattice oxygen in WO₃) can be replenished to recover activity (Fig. 2c and

Supplementary Fig. 21). As such, a durable photochemical CH₄ conversion process can be established by recycling the photochemical CH₄ conversion and air recovery. The durability measurement indicates that the performance of PdO/Pd−WO₃-2 is well maintained for five cyclic tests with each cycle lasting 5 h in such a recycling system (Supplementary Fig. 22). Moreover, the leaching out of Pd during the cyclic tests is also negligible according to the results of mass spectrometry (Supplementary Fig. 23).

## Reaction intermediates detection

The information gleaned above has recognized the promising performance for the conversion of CH₄ to CH₃COOH by modulating Pd/PdO heterostructure. Naturally, a question arises how CH₄ evolves into CH₃COOH over PdO/Pd−WO₃ nanocomposites without additional carbon sources. To this end, we investigate the reaction intermediates over the nanocomposites during the photochemical CH₄ conversion process. Figure 3a shows the in situ diffuse reflectance-infrared Fourier-transform spectra (DRIFTS) for light-driven CH₄ conversion over Pd/WO₃ sample. Upon light irradiation, apart from the peaks at 1305 and 3015 cm⁻¹ corresponding to C−H deformation vibration of CH₄, the peak at 1439 cm⁻¹ for CH₂/CH₃ deformation vibration appears gradually, indicating the CH₄ dissociation over the sample[35,36]. Moreover, the significant growth of vibrational peak at 1061 cm⁻¹ and bands at 2927 and 2963 cm⁻¹, corresponding to the methoxy and C−H stretching vibrations in CH₃OH product, can be attributed to the CH₄ activation in the presence of ·OH[37]. In sharp contrast, PdO/Pd−WO₃

nanocomposites that can produce $CH_3COOH$ through light-driven $CH_4$ conversion exhibit different behavior in DRIFTS (Fig. 3b). In addition to the vibration signals of $CH_3OH$ observed over $Pd/WO_3$, the additional vibrational modes of $C=O$ (1654 cm$^{-1}$), C–O (979 cm$^{-1}$), C–C (867 cm$^{-1}$) and C–H (2858 cm$^{-1}$) stretching vibrations can be monitored for the formation of $CH_3COOH$ over $PdO/Pd$-$WO_3$-2[38,39]. Notably, a broad peak at 2060 cm$^{-1}$ in Fig. 3b is observed with the light irradiation proceeding, which can be assigned to the adsorbed *CO on Pd site (Supplementary Fig. 24)[40]. The vibration signals of $C=O$ and *CO only appear with the existence of PdO species, implying that the synergistic effect of Pd/PdO heterostructure in the nanocomposite can facilitate the $CH_3COOH$ production with *CO as an intermediate.

To further understand the process with elemental information, the surface carbon and oxygen species are also monitored by in situ near ambient pressure XPS (NAP-XPS) characterization. As shown in Fig. 3c, after introducing the reactant into NPA-XPS chamber, the peak of gas-phase $CH_4$ (287.0 eV) is observed in the high-resolution C 1$s$ XPS spectrum (Supplementary Fig. 25). Upon light irradiation, three C 1$s$ peaks of surface ·$CH_x$ (285.5 eV), C–O (286.1 eV) and COO (289.1 eV) species appear and increase with the time evolution[41–43]. Meanwhile, the formation of oxygenates from $CH_4$ oxidation is also verified by collecting the O 1$s$ spectra in NAP-XPS studies. Apart from the peak of lattice oxygen in sample (**O1**, 530 eV), the featured peaks of hydroxyl (**O2**, 531.1 eV), C–O (**O3**, 531.8 eV), adsorbed $H_2O$ (**O4**, 532.5 eV) and $C=O$ (**O5**, 533.2 eV) species are resolved after light illumination (Fig. 3d)[44–46]. Of note, although the signals of *CO have been detected by in situ DRIFT and NAP-XPS measurements, gaseous CO is not observed as a product. Indeed, previous work has demonstrated that the adsorption of CO on PdO site is extremely strong so that *CO would be coupled with other intermediates before desorption[47]. Apparently, the surface ·$CH_x$, C–O and $C=O$ species are corroborated with the observation of in situ DRIFTS spectra. This indicates that the co-adsorption of $CH_4$ and $H_2O$ over Pd/PdO heterostructure can produce various surface carbonaceous intermediates including methyl and carbonyl species and further generate liquid oxygenates.

## Mechanistic study

As revealed by in situ characterizations, the carbonyl species is the key intermediate for the conversion of $CH_4$ to $CH_3COOH$, which is closely correlated with the presence of PdO in the prepared nanocomposite. The case of $Pd/WO_3$ reveals that the generation of ·OH radicals alone cannot lead to the formation of $C=O$ in the absence of PdO species (Fig. 3a), implying that the oxygen in $C=O$ is most likely derived from PdO in the nanocomposite. To further trace the oxygen source of carbonyl intermediate in $CH_3COOH$ production, we prepare the $^{18}O$-labeled PdO-modified nanocomposite (denoted as $Pd^{18}O/Pd$-$WO_3$-2) by annealing pristine $Pd/WO_3$ in $^{18}O_2$ atmosphere. Subsequently, the light-driven $CH_4$ oxidation is performed over $Pd^{18}O/Pd$-$WO_3$-2 and the products are analyzed by gas chromatography–mass spectrometry (GC–MS), in reference to $Pd^{16}O/Pd$-$WO_3$-2. In contrast to the case of $Pd^{16}O/Pd$-$WO_3$-2, the peaks at $m/z = 45$ and 47 by $Pd^{18}O/Pd$-$WO_3$-2 can be ascribed to $CH_3C^{18}O^+$ and $^+C^{18}OOH$ (Fig. 4a), indicating that the O atom in *CO is derived from the lattice oxygen of $Pd^{18}O$ in sample. As a result, the peak at $m/z = 62$ for $CH_3C^{18}OOH$ can be detected.

To further understand the working mechanism of PdO in the production of $CH_3COOH$, we quantitatively establish the relation between PdO consumption and $CH_3COOH$ production. With this purpose, the $H_2$-temperature-programmed reduction ($H_2$-TPR) is employed to determine the content of PdO in $PdO/Pd$-$WO_3$-$x$ nanocomposites (Supplementary Fig. 26). As revealed in Fig. 4b, the amounts of PdO in samples are very close to the $CH_3COOH$ yield at low PdO contents (i.e., $PdO/Pd$-$WO_3$-1 and $PdO/Pd$-$WO_3$-2). Given that the PdO is completely consumed after photochemical tests, this further confirms that the lattice oxygen of PdO solely contributes to

the formation of $C=O$ in $CH_3COOH$ during photochemical $CH_4$ oxidation process. However, excessive PdO in nanocomposites will lower the content of metallic Pd to form $PdO$-$WO_3$ interface, which suppresses charge separation and reduces performance[48–50]. In this case, the low $CH_4$ photooxidation performance allows most lattice oxygen atoms in PdO to remain in the samples (i.e., $PdO/Pd$-$WO_3$-3 to $PdO/Pd$-$WO_3$-5 in Fig. 4b). It is worth pointing out that the formation of Pd-PdO interface in nanocomposite is critical for $CH_3COOH$ generation. Our control experiments indicate that $CH_3COOH$ cannot be produced through simply mixing $Pd/WO_3$ with PdO, suggesting that Pd-PdO interface is a key factor for $CH_3COOH$ generation (Supplementary Fig. 27). Moreover, the $CH_3COOH$ production depends on the structural character of Pd/PdO heterointerface (Supplementary Figs. 28 and 29), corroborating the importance of Pd/PdO interface quality to $CH_3COOH$ synthesis.

Taken together, the experimental results have revealed the critical role of Pd/PdO heterointerface in $CH_4$-to-$CH_3COOH$ conversion. Figure 4c illustrates the proposed reaction pathway. $CH_4$ prefers to be activated at Pd site in the presence of ·OH and form $Pd$-$CH_3$ intermediate (Supplementary Fig. 30). The methyl species can be gradually converted to Pd–CO intermediate through the combination with O atom from PdO and the dehydrogenation by ·OH. Subsequently, the C–C coupling between carbonyl and methyl species generates the Pd–COCH$_3$ intermediate at Pd–PdO interface, and the further hydrolysis of Pd–COCH$_3$ gives $CH_3COOH$ as a product[51–53]. Of note, W sites are also active for *$CH_3$ generation by directly oxidizing $CH_4$ on $WO_3$. However, the *$CH_3$ formed on $WO_3$ can hardly approach the *CO on PdO so that the surplus *$CH_3$ species would evolve into $C_1$ oxygenates[54,55].

From the working mechanism of Pd/PdO heterostructure, we can now understand that the complete consumption of lattice oxygen in PdO during light-driven $CH_3COOH$ production, in the case of $PdO/Pd$-$WO_3$-2, will inevitably lead to the significant performance decay for $CH_3COOH$ production after the reaction (Supplementary Fig. 31). In comparison, the $CH_3OH$ production is not significantly affected by the oxygen loss in PdO. It is worth noting that the evolution of $WO_3$ in photochemical $CH_4$ conversion is also a factor for performance decay. Given that no $H_2$ is detected in the photochemical process, the $WO_3$ is inevitably reduced by photo-induced electrons, which is accompanied with gradual lattice oxygen loss, also causing performance decay (Supplementary Fig. 20). Nevertheless, the amount of lost oxygen atoms in $WO_3$ is determined to be 1.28% through the calculation based on the demand of ·OH radical production, which is negligible as compared with the consumption of PdO (taking $PdO/Pd$-$WO_3$-2 as example). Furthermore, the lost lattice oxygen in $WO_3$ can be replenished together with that in PdO during the regeneration process, recovering photochemical activity.

Following the mechanistic studies, our investigation on photochemical $CH_4$ conversion in gas-solid phase indicates that the generated *$CH_3$ may undergo self-coupling to produce $C_2H_6$ as the primary product (Supplementary Fig. 32). For this reason, the controllable utilization of *$CH_3$ in solution and gas phases is of great importance to further improve the production rate and selectivity for $CH_3COOH$.

## Photochemical flow synthesis of $CH_3COOH$

The key to controllable *$CH_3$ utilization is the efficient methyl–carbonyl coupling. Certainly, such an efficient coupling should be based on the supply of sufficient *CO species. Our control experiments show that the addition of CO to the reaction system using $Pd/WO_3$ nanocomposite, in the absence of PdO, can deliver similar $CH_3COOH$ production (Supplementary Fig. 33). In comparison, the addition of methanol does not obviously promote the $CH_3COOH$ production over $PdO/Pd$-$WO_3$-2 nanocomposite, suggesting that $CH_3COOH$ is not the product primarily from methanol carbonylation (Supplementary Fig. 34). The results provide us the clues for enhancing $CH_3COOH$

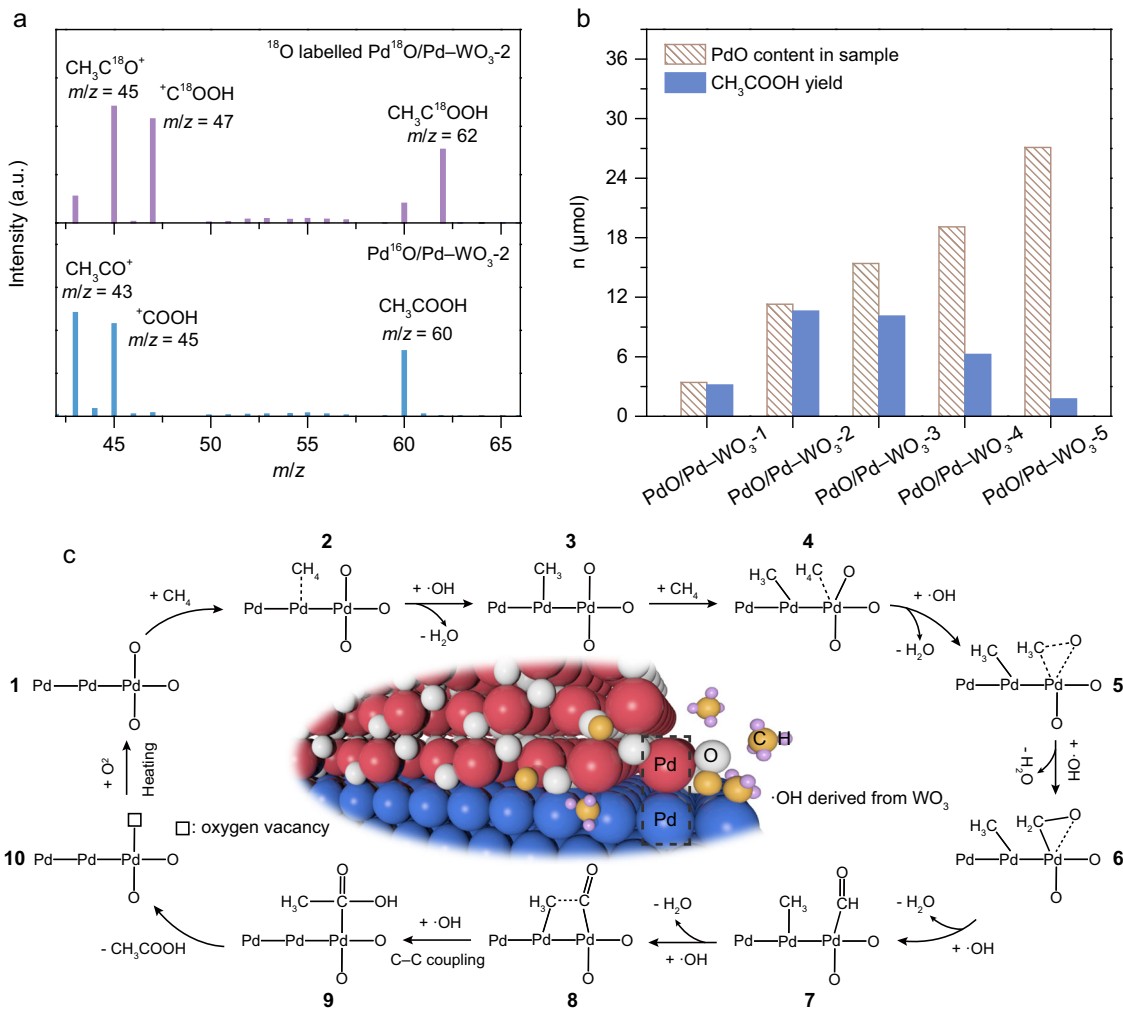

**Fig. 4 | Mechanism for photochemical CH₄ to CH₃COOH conversion. a** Mass spectra of CH₃COOH product using Pd¹⁶O/Pd–WO₃-2 and ¹⁸O-labeled Pd¹⁸O/Pd–WO₃-2 nanocomposite. **b** The comparison of PdO contents in samples (50 mg) and the CH₃COOH yields of the corresponding samples. **c** Schematic illustration for photochemical conversion of CH₄ to CH₃COOH over Pd/PdO heterointerface in the presence of ·OH radicals. The numbers represent the reaction steps.

production—the cascade reaction between *CO and *CH₃ on nanocomposite in continuous reaction channels that can promote the utilization of PdO and *CH₃.

To this end, we design a photochemical flow reaction device with arc-shaped flow channels to further enhance the performance of CH₃COOH production (Supplementary Fig. 35). In this design, the *CH₃ species that have not coupled with *CO can migrate along the sample to further react with the adsorbed *CO or even evolve into *CO on the downstream PdO sites, promoting the conversion of CH₄ to CH₃COOH. Specifically, CH₄ and H₂O are premixed to form the monodispersed gas bubbles, which are then pumped into the flow reactor to generate gas-liquid-solid contact in channels (Fig. 5a). Benefitting from the flowing reactants and three-phase interface between CH₄, H₂O and sample (Fig. 5b, c), the generated *CH₃ in solution from gas-solid phase CH₄ oxidation can be rapidly captured by *CO on sample layer to realize continuous synthesis of CH₃COOH. As such, the remarkable selectivity of 91.6% and production rate of 90.7 µmol g⁻¹ h⁻¹ are achieved for CH₃COOH production over PdO/Pd–WO₃-2 nanocomposite (Fig. 5d). As normalized to the Pd loading weight, the production rate reaches 1.5 mmol g$_{Pd}$⁻¹, which exceeds the performance of existing photocatalysts for oxygenates production under mild condition (Supplementary Table 2). Furthermore, the integration of photochemical CH₄ conversion with regeneration

process also demonstrates the reproducibility and durability of the flow reaction device (Fig. 5e).

## Discussion

We have demonstrated a direct light-driven synthesis of CH₃COOH solely from CH₄ on PdO/Pd–WO₃ heterointerface nanocomposite, by controlling carbonyl intermediate formation and methyl–carbonyl coupling. As revealed by solid evidence from in situ characterizations, the PdO species can convert CH₄ into carbonyl intermediate, holding the key to CH₃COOH production. Our isotope labeling experiments indicate that the oxygen atom in carbonyl intermediate is derived from the lattice oxygen of PdO in nanocomposite, providing important information for establishing a conversion–regeneration process toward long-term recyclability. Leveraging our understanding on CH₄-to-CH₃COOH conversion pathway, we have designed a photochemical flow reaction device enabling cascade reactions to enhance the efficiency and selectivity of acetic acid production. As a result, the approach achieves the impressive production rate of 1.5 mmol g$_{Pd}$⁻¹ h⁻¹ and selectivity of 91.6% toward CH₃COOH. This work highlights the importance of rational heterostructure engineering to controlling intermediates evolution, and provides new insights for selective C₂₊ oxygenates synthesis using methane as resource under mild conditions.

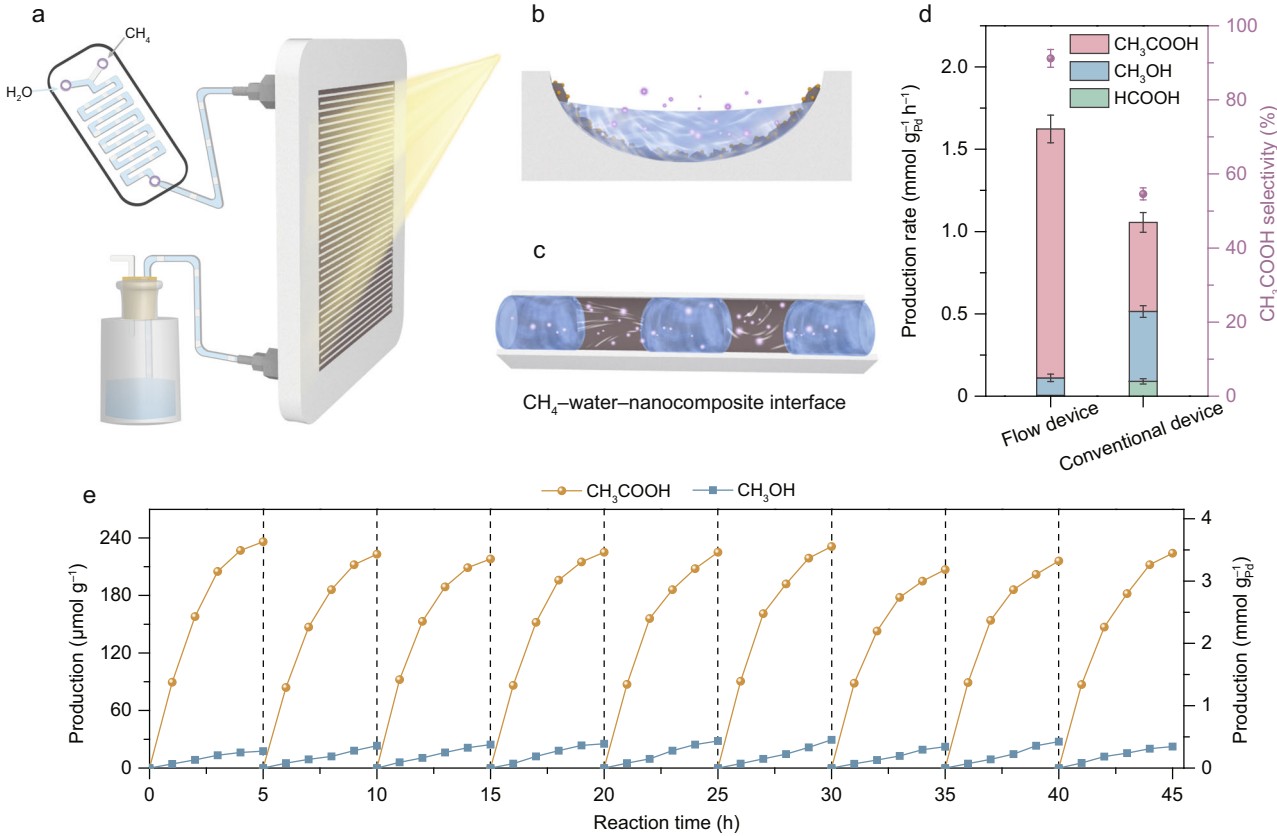

**Fig. 5 | Photochemical flow synthesis of CH₃COOH from CH₄. a** Schematic illustration of photochemical flow reaction device, including reactants supplier, homemade reactor and products collector. **b, c** Side (**b**) and top (**c**) views of the arc-shaped flow channel in homemade reactor and the three-phase contact between CH₄, H₂O and sample. The purple, blue and brown colors represent CH₄, H₂O and nanocomposite, respectively. **d** Production rate and selectivity of light-driven CH₄ conversion toward CH₃COOH over PdO/Pd–WO₃-2 nanocomposite using the flow reaction device or conventional device for the first 3 h. **e** Reaction-regeneration cycles on PdO/Pd–WO₃-2 sample by employing the flow reaction device. The error bars represent the standard deviation of the experiments.

## Methods

### Chemicals

Sodium tungstate dehydrate ($Na_2WO_4 \cdot 2H_2O$, 99.5%), citric acid (CA, 99.5%) and palladium chloride ($PdCl_2$, 98% metals basis) were purchased from Aladdin. Sodium borohydride ($NaBH_4$, 98%) was obtained from Sigma-Aldrich. Hydrochloric acid (HCl, 36 ~ 38%) and ascorbic acid (AA, 99.7%) were purchased from Sinopharm Chemical Reagent Co., Ltd. The water used in all experiments was deionized. All of the chemical reagents were used as received without further purification.

### Materials preparation

$WO_3$ nanosheets were prepared by a two-step process. Typically, 1 mmol $Na_2WO_4 \cdot 2H_2O$ and 1.5 mmol citric acid were dissolved in 30 mL $H_2O$ to form a transparent solution. Then 3 mL of HCl solution (6 M) was added into the solution with vigorous stirring for 30 min. The mixture was transferred into a 50 mL Teflon-lined autoclave and heated at 120 °C for 24 h. The resulting precursor of $WO_3 \cdot H_2O$ nanosheets was centrifuged and washed with water several times, and dried in a vacuum oven. The $WO_3$ nanosheets were obtained by calcinating the collected solid in air at 400 °C for 2 h. For the synthesis of Pd/$WO_3$ nanocomposites, 60 mg $WO_3$ nanosheets were dispersed in 30 mL water to form a homogeneous suspension. Then 6.5 mg of $PdCl_2$ was dissolved in 0.5 mL HCl solution (10 mM), which was added into the $WO_3$ suspension and further reacted with 5 mg of $NaBH_4$. The slurry was washed with water three times and dried in a vacuum oven, producing Pd/$WO_3$ sample. The PdO/Pd–$WO_3$-1, PdO/Pd–$WO_3$-2 and PdO/Pd–$WO_3$-3 nanocomposites were obtained by calcinating the Pd/$WO_3$ sample at 120 °C, 200 °C and 260 °C for 5 h, respectively, with a heating rate of 1 °C min⁻¹ in air. The PdO/Pd–$WO_3$-4 and PdO/Pd–$WO_3$-5 nanocomposites were obtained by calcinating the Pd/$WO_3$ sample at 350 °C and 450 °C for 3 h and 2 h, respectively, with a heating rate of 2 °C min⁻¹ in air.

### Characterization

Powder XRD patterns were measured by Philips X'Pert Pro Super X-ray diffractometer with Cu-Kα radiation ($\lambda = 1.54178$ Å). XPS characterizations of the prepared samples were carried out on JPS-9010MC (JEOL, Japan) with a hemispherical electron energy analyzer (1486 eV Al Kα radiation). TEM images were taken on a Hitachi Model H-7700 microscope at 100 kV. HRTEM images were taken on a JEOL JEM-2100 field-emission higher-resolution transmission electron microscope at 200 kV. The aberration-corrected HAADF-STEM images and EELS analysis were collected on the JEOL ARM-200F field-emission transmission electron microscope operated at 200 kV. EPR spectra for radical detection were obtained on the JEOL JES-FA200 spectrometer.

### Photochemical CH₄ conversion measurement

In a typical test, 10.0 mg of sample was dispersed in 10 mL water and added into a 30 mL custom-made quartz tube reactor. The light-driven CH₄ conversion experiments were carried out in pure CH₄ atmosphere (0.1 MPa) at room temperature. The reactor was irradiated by a 300 W xenon lamp (PLS-SXE300, Perfect light) with light intensity of 200 mW cm⁻². The gas products were quantified by a gas chromatograph (GC, 7890B, Ar carrier, Agilent) equipped with thermal conductivity detector (TCD) and flame ionization detector (FID). Another GC (Techcomp GC-7900, China) equipped with a TDX-01 packed

column was employed to measure the amounts of CO and $CO_2$. The liquid products were quantified by $^1H$ NMR (Bruker Avance, 600 MHz) with a water suppression pulse sequence. A certain concentration of dimethyl sulfoxide (DMSO) solution was used as external standard to calibrate the liquid products. The trapping experiments were performed by adding 1 mM $K_2Cr_2O_7$, $Na_2C_2O_4$ and salicylic acid into the reaction solution as photo-induced electron, hole and ·OH scavengers, respectively.

For using the designed photochemical flow device, 100 mg of sample was loaded on the channel of the homemade flow reactor. The reactor was clamped with mould and quartz plate. The reactants of $CH_4$ and $H_2O$ were premixed by the microfluidic device to form the monodisperse gas and bubble, which were then pumped into the reactor for photochemical conversion under 300 mW cm$^{-1}$ of light irradiation. The liquid products were received in bottle. For recovering the photochemical performance, the sample was calcinated at 230 °C for 3 h with a heating rate of 1 °C min$^{-1}$ in air.

### Isotope-labeling experiments
The isotope-labeling experiments were performed by using pure $^{13}CH_4$ and $^{12}CH_4$ as feeding gas. The liquid products were detected by $^{13}C$ NMR. To trace the oxygen atom of $CH_3COOH$, the PdO species in nanocomposite was generated by calcinating Pd/$WO_3$ nanocomposite in $^{18}O_2$ atmosphere at 200 °C for 8 h to label the oxygen atoms in PdO. The photochemical tests were performed in the homemade flow reaction device for maximizing $CH_3COOH$ yield. The $CH_3COOH$ product was concentrated and then analyzed by GC−MS (7890 A and 5975 C, He carrier, Agilent).

### Photocurrent measurements
The photocurrent tests of the prepared samples were conducted on CHI 660D electrochemical workstation (CH Instruments) with three-electrode system under light or dark condition. Typically, 5.0 mg of material was dispersed in 500 μL of ethanol/water mixture (4:1, v/v) and then dropped onto a 1 × 3 cm fluorine-doped tin oxide (FTO)-coated glass for work electrode preparation. The Pt foil and saturated Ag/AgCl electrode were employed as counter and reference electrode, respectively. The measurements were performed using 0.5 M $Na_2SO_4$ aqueous solution as electrolyte. The photocurrent responses of the photoelectrodes (i.e., I−t curves) were collected by measuring the photocurrent densities under chopped light irradiation (light on/off cycles: 10 s) at a bias potential of 0.8 V vs. Ag/AgCl.

### Detection of hydroxyl and methyl radicals
Briefly, the sample and 5,5-dimethyl-1-pyrroline N-oxide (DMPO) were dispersed in ice-bath water. The mixture was vigorously shaken and irradiated by using a 500 W xenon lamp, and then analyzed by EPR spectroscopy. Methyl radical was trapped by the same procedure under pure $CH_4$ in the reaction system.

### In situ DRIFTS for photochemical $CH_4$ conversion
In situ DRIFTS measurements were performed at BL01B in the NSRL in Hefei, China. The spectra were collected by using a Bruker IFS 66 v Fourier-transform spectrometer equipped with Harrick diffuse reflectance accessory with ZnSe and quartz window. Each spectrum was recorded by averaging 128 scans at a resolution of 2 cm$^{-1}$. After sample loading, pure $CH_4$ (99.999%) and water vapor were introduced into the chamber for background spectra collection. After that, the system was exposed to light irradiation and the spectra were collected when the irradiation times were 1, 5, 10, 20 and 30 min, respectively.

### In situ NAP-XPS measurement for photochemical $CH_4$ conversion
In situ NAP-XPS measurements were carried at the beamline BL02B1 of SSRF under light irradiation or dark condition. The sample was

dropped onto a silicon wafer and subsequently cleaned by Ar plasmon for 10 min to remove the surface agent on sample. The prepared sample was stored in the vacuum before the measurement. The XPS spectra were recorded under dark condition firstly. After that, the reactant was sequentially introduced into the analysis chamber with the partial pressure up to 45 Pa. Subsequently, the in situ NAP-XPS spectra were collected under 365 nm LED light irradiation.

## Data availability
The authors declare that all data supporting the findings of this study are available in the article and its Supplementary Information. Source data are provided with this paper. Figure 1g, Fig. 2a−c, Fig. 3a−d, Fig. 4a−b, Fig. 5d−e, Fig. S4, Fig. S6, Fig. S8, Fig. S9, Fig. S11, Fig. S17, Fig. S19, Fig. S22, Fig. S24. Additional data are available from the corresponding author upon reasonable request. Source data are provided with this paper.

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

## Acknowledgements

This work was financially supported by the National Key R&D Program of China (2020YFA0406103), NSFC (22122506, 22232003, 22075267, 21725102, 91961106), Strategic Priority Research Program of the CAS (XDPB14), Anhui Provincial Natural Science Foundation (2008085J05), Youth Innovation Promotion Association of CAS (2019444), Open Funding Project of National Key Laboratory of Human Factors Engineering (SYFD062010K), and Fundamental Research Funds for the Central Universities (KY2140000031, 20720220007, WK2060000039).

The in situ DRIFTS measurements were performed at beamline BL01B in the NSRL. NAP-XPS measurements were performed at the beamline BL02B1 of SSRF supported by National Natural Science Foundation of China under contract no. 11227902. The authors thank the support from USTC Center for Micro- and Nanoscale Research and Fabrication.

## Author contributions

R.L. and Y.X. supervised the projects. W.Z., R.L. and Y.X. conceived the idea for this work. W.Z. prepared the photocatalysts, carried out catalytic measurements and in situ experiments. W.Z., D.X., Y.C., A.C. and Y.J. contributed to the characterization. W.Z., R.L. and Y.X. analyzed the data. W.Z., Z.Z., H.Z. and Z.L. contributed to the NAP-XPS measurements. W.Z. and H.L. contributed to the DRIFTS measurements. W.Z., R.L. and Y.X. wrote the manuscript. All the authors contributed to the interpretation of the data and preparation of the manuscript.

## Competing interests

The authors declare no competing interests.
