## [Peer Review File · Nature Communications]

Light-driven flow synthesis of acetic acid from methane with chemical loopingREVIEWER COMMENTS

Reviewer #1 (Remarks to the Author):

Prof. Xiong and coworkers reported PdO/Pd-WO₃ as a ternary photocatalyst for the conversion of CH₄ into CH₃COOH without additional oxidants in this manuscript. It was found that the optimal content of PdO on PdO/Pd-WO₃ heterointerface catalysts was crucial for the selectivity and production rate toward the photocatalytic conversion of CH₄ into CH₃COOH. Overall, this work seems interesting and well investigated, but some critical issues should be addressed completely before making a final decision for publication in Nature Communications.

1. As stated, the PdO/Pd-WO₃ catalyst was consumed during the conversion reaction of CH₄ into CH₃COOH and required a regeneration process for its further use in the next round of reaction. Such a material should have been claimed as a reagent but not as a catalyst since typical catalysts are not consumed and do not require additional regeneration processes.
2. It is not clear which one is a predominant active species, Pd(0) or Pd(2+), on PdO/Pd-WO₃ for CH₄ activation into *CH₃? In the proposed mechanism (Fig. 4c), both seem to be active, however, there is no experimental evidence in the current work.
3. In addition, according to the proposed mechanism, the photogenerated holes of WO₃ on PdO/Pd-WO₃ induced the oxidation of H₂O to form hydroxyl radicals. However, the photoexcited electrons of WO₃ are missing in the proposed mechanism, which should be clarified via some experimental results, including the use of electron and hole scavengers.
4. The authors claimed that the lower content of metallic Pd [Pd(0)] on PdO/Pd-WO₃ suppressed charge separation, leading to reduction in its photocatalytic performance. However, there is no experimental evidence for this claim, so the lifetime of the excited electrons for PdO/Pd-WO₃-1, 2, 3, 4, and 5 should be measured.
5. The lattice oxygen of PdO/Pd-WO₃ was consumed during the conversion reaction of CH₄ into CH₃COOH, which could make Pd atoms also leaching out into the reaction solution. After the first round of reaction, the Pd content of the reaction solution as well as PdO/Pd-WO₃ should be measured by ICP. Besides, after regeneration of PdO/Pd-WO₃, the total Pd content should be confirmed and compared with the initial one.
6. How about the performance of a physical mixture of Pd-WO₃ and PdO nanoparticles for the conversion of CH₄ into CH₃COOH under light irradiation?
7. The details of experimental procedures should be supplemented for others to repeat them. There is no experimental procedure for the regeneration of the used PdO/Pd-WO₃ in the current manuscript.

8. The authors should do comprehensive literature search for the synthesis of PdO/Pd-WO₄ hybrid structures to emphasize the novelty and significance of this work in terms of materials chemistry.

Reviewer #2 (Remarks to the Author):

This article by Xiong and coworkers reported a direct synthesis of CH₃COOH solely from CH₄ via photocatalysis without additional reagents. High selectivity and yield toward CH₃COOH are achieved in a photocatalytic flow reactor, along with the novel MVK oxidation mechanism for forming of CO*, are excellent features of this work. The study is well conducted, the SI is complete, and the results clearly presented. I think this work deserves to be considered for publication in Nature Comm. However, some important mechanistic concerns should first be addressed before acceptance of this work.

1. CO gas was not detected in the products in this work, but the CO* was confirmed by in situ DRIFTS measurements. Why the CO* could not be further oxidated to CO₂?

2. P9, line153-155, 'with the further increase of PdO content, the Pd-WO₃ interface is gradually diminished, which reduces photo-induced charge separation efficiency and substantially suppresses photocatalytic performance', the authors thought that the photo-induced charge separation efficiency decreased with the increase of PdO content. However, this point should be proved by experimental data, such as photocurrent response data.

3. In Supplementary Fig. 21, the most of CH₄ could not be oxidated to CO* and then to form CH₃COOH in the in gas-solid phase, but transformed to C₂H₆. Does it mean that more H₂O is helpful for forming of CO*? Why?

4. In this work, the ·OH radical derived from water oxidation by photo-induced hole, how about the role of the photo-induced electron?

Reviewer #3 (Remarks to the Author):

The manuscript "Photocatalytic flow synthesis of acetic acid solely from methane with high selectivity by heterointerface catalyst" demonstrates the synthesis of acetic acid directly from methane using Pd-PdO supported over WO₃. It shows high selectivity to acetic acid, especially in flow conditions, however, the productivity is limited due to the looping process of methane oxidation by oxygen of Pd. The research is well performed and the manuscript clearly represents the discovery. I would recommend it for publication after clarification of several points:

1. First of all, I would not call it "catalysis" because the catalyst cannot be consumed during the reaction as in this case. Thus, I would use the term "photochemical looping".

2. The effect of the calcination temperature on the Pd/PdO ratio and acetic acid yield looks reasonable, however, it seems strange to me that, for example, samples 2 and 4 containing comparable amounts of O according to TPR have so high differences in the production rate of acetic acid. In fact, PdO is a very

flexible material and usually can be easily oxidized and reduced. I would expect an increase in the size of metal nanoparticles or interaction of Pd with WO₃ at high calcination temperatures could be also important. I would recommend to perform the reduction of the sample calcined at high temperature (for example 5) and oxidation at low temperature (200 C) to check if it brings it to the performance of sample 2. If not the changes are not reversible and have structural character and it would be important to identify them.

3. The authors do not talk a lot about the role of WO₃ in the reaction. What would happen if WO₃ is substituted by TiO₂? Is it important for the stabilization of Pd-PdO interface? It could be important to clarify the effect of semiconductor and perform its characterization (band gap...).

4. Is there hydrogen generation during the reaction? It could be formed during methane and water splitting.

5. There is a peak of CO over Pd at 2060 cm⁻¹ which is usually assigned to CO adsorption over edges. It could be interesting to perform FTIR CO over all samples to check the effect of calcination temperature on the state and morphology of surface Pd atoms.

6. The opposite trends of methanol and acetic acid production could be explained by the intermediate synthesis of methylformate which can be isomerized to acetic acid over Pd catalyst (<https://doi.org/10.1002/cssc.202201006>). It would explain why the selectivity is high in flow conditions at deeper conversion. The migration of CH₃ radical in the flow seems strange to me because the lifetime of radical should be low. It could be interesting to try PdO-Pd/WO₃ for the oxidation of methanol – probably it could be converted to acetic acid.

7. Recently the photocatalytic synthesis of acetic acid by carbonylation of methane has been published in *J. Am. Chem. Soc.* 2023, 145, 2, 1185–1193. It would be important to refer to this work.

Point-by-point response to the reviewers' comments

Reviewer #1 (Remarks to the Author):

Prof. Xiong and coworkers reported PdO/Pd-WO₃ as a ternary photocatalyst for the conversion of CH₄ into CH₃COOH without additional oxidants in this manuscript. It was found that the optimal content of PdO on PdO/Pd-WO₃ heterointerface catalysts was crucial for the selectivity and production rate toward the photocatalytic conversion of CH₄ into CH₃COOH. Overall, this work seems interesting and well investigated, but some critical issues should be addressed completely before making a final decision for publication in Nature Communications.

Author Response: We really appreciate the referee's positive evaluation for our work, and are grateful to the referee for his/her insightful suggestions to help us further improve the quality of our manuscript.

1. As stated, the PdO/Pd-WO₃ catalyst was consumed during the conversion reaction of CH₄ into CH₃COOH and required a regeneration process for its further use in the next round of reaction. Such a material should have been claimed as a reagent but not as a catalyst since typical catalysts are not consumed and do not require additional regeneration processes.

Author Response: We thank the referee for his/her insightful suggestion. We agree that the *CO intermediate and CH₃COOH are both derived from the chemical looping of PdO in our materials. According to the suggestion, we have revised the title and corresponding description in the manuscript.

*2. It is not clear which one is a predominant active species, Pd(0) or Pd(2+), on PdO/Pd-WO₃ for CH₄ activation into *CH₃? In the proposed mechanism (Fig. 4c), both seem to be active, however, there is no experimental evidence in the current work.*

Author Response: We thank the referee for his/her thoughtful suggestion. We agree that the Pd(0) and Pd(2+) sites both exhibit activity for CH₄ activation into *CH₃ no matter in our work or previous works, but yield different final products. Generally, the Pd(0) can generate and stabilize *CH₃ for *CH₃ coupling (ACS Catal. 2021, 11, 13768–13781), while CH₄ undergoes carbonylation and combustion (CH₄ → CH₃O → ... → CO₂) over Pd(2+) site through Mar–van Krevelen mechanism (Angew. Chem. Int. Ed. 2021, 60, 18552–18556). To further validate their role in PdO/Pd–WO₃-2, we have performed the light-driven nonoxidative coupling of methane over the three typical samples. As shown in the newly added Supplementary Fig. 30, the catalysts exhibit the order of Pd/WO₃ > PdO/Pd–WO₃-2 > PdO/Pd–WO₃-5 for C₂H₆ production. In

particular, abundant CO₂ is detected over the PdO/Pd–WO₃-5 sample while negligible C₂H₆ is observed, implying that the Pd(0) on PdO/Pd–WO₃ is the predominant active species for *CH₃ generation as compared with Pd(2+). We have added the related data and discussion into the revised manuscript.

Supplementary Fig. 30 The light-driven nonoxidative coupling of methane (NOCM) performance over the typical Pd/WO₃, PdO/Pd–WO₃-2 and PdO/Pd–WO₃-5 samples in 0.1 MPa CH₄.

3. In addition, according to the proposed mechanism, the photogenerated holes of WO₃ on PdO/Pd-WO₃ induced the oxidation of H₂O to form hydroxyl radicals. However, the photoexcited electrons of WO₃ are missing in the proposed mechanism, which should be clarified via some experimental results, including the use of electron and hole scavengers.

Author Response: We thank the referee for his/her valuable suggestion. According to the suggestion, we have performed the control experiments using electron, hole and hydroxyl radical scavengers. As shown in the newly added Supplementary Fig. 14, introducing Cr₂O₇²⁻ as electron scavenger does not affect the production rates. However, the addition of C₂O₄²⁻ as hole scavenger or salicylic acid as OH scavenger reduces the production rates significantly, indicating that the CH₄ photooxidation process is triggered by OH radicals.

Supplementary Fig. 14 Production rates of photochemical CH₄ conversion over PdO/Pd-WO₃-2 with 1 mM Cr₂O₇²⁻, C₂O₄²⁻ and salicylic acid added as photoinduced electron, hole and OH scavengers, respectively.

As shown in the band structures of the prepared WO₃-based materials (the newly added Supplementary Figs. 19 and 20), the photoexcited electrons of the materials cannot reduce H₂O to evolve H₂. Based on the band structure, the WO₃ is inevitably reduced by photo-induced electrons, which is accompanied with gradual lattice oxygen loss, also causing performance decay. The amount of lost oxygen atoms in WO₃ is determined to be 1.28% through the calculation based on the demand of OH radical production, which is negligible as compare with the consumption of PdO (taking PdO/Pd-WO₃-2 as an example). Therefore, the performance decay by the consumption of PdO is more significant than that by the lattice oxygen loss in WO₃, especially for CH₃COOH production. More importantly, based on the requirement of PdO regeneration process, the lattice oxygen in WO₃ is also replenished together with PdO. We have now included the related data and discussion in the revised manuscript.

Supplementary Fig. 19 UV-vis diffuse reflectance spectra of the as-prepared samples.

Supplementary Fig. 20 (a) Valence band spectra and (b) secondary electron cutoff for the as-prepared samples. The excitation photon energy is 40 eV. (c) The electronic band structures of the prepared samples.

4. The authors claimed that the lower content of metallic Pd [Pd(0)] on PdO/Pd-WO₃ suppressed charge separation, leading to reduction in its photocatalytic performance. However, there is no experimental evidence for this claim, so the lifetime of the excited electrons for PdO/Pd-WO₃-1, 2, 3, 4, and 5 should be measured.

Author Response: We thank the referee for his/her thoughtful suggestion. According to the suggestion, the time-resolved PL (TRPL) decay spectra of the as-prepared samples have been collected (the newly added Supplementary Fig. 10). The average PL lifetimes are shortened from 0.95 ns for WO₃ to 0.62 ns for PdO/Pd-WO₃-1, which is attributed to the electron transfer channel from WO₃ to metallic Pd. Incorporating PdO into nanocomposites with appropriate content maintains the PL lifetimes (PdO/Pd-WO₃-2 and PdO/Pd-WO₃-3 samples). Nevertheless, the excessive PdO content shuts the channel and thus the PL lifetimes are prolonged (PdO/Pd-WO₃-4 and PdO/Pd-WO₃-5 samples), indicating that the extra PdO hinders the Schottky contact between Pd and WO₃ and is detrimental to charge separation. We have now included the related data

and discussion in the revised manuscript.

Supplementary Fig. 10 Time-resolved photoluminescence (TRPL) decay of the as-prepared samples.

5. The lattice oxygen of PdO/Pd-WO₃ was consumed during the conversion reaction of CH₄ into CH₃COOH, which could make Pd atoms also leaching out into the reaction solution. After the first round of reaction, the Pd content of the reaction solution as well as PdO/Pd-WO₃ should be measured by ICP. Besides, after regeneration of PdO/Pd-WO₃, the total Pd content should be confirmed and compared with the initial one.

Author Response: We thank the referee for his/her thoughtful suggestion. According to the suggestion, we have measured the residual Pd content in reaction solution using ICP-MS. As presented in the newly added Supplementary Fig. 23, the Pd²⁺ can only be detected in the first round of reaction solution, corresponding to 0.16% Pd loss in the first cycle. During the measured three cyclic tests, the photochemical performance is well maintained, suggesting that the Pd atoms are stable in the PdO/Pd-WO₃ heterointerface. The total Pd content is measured by ICP-OES, showing that the Pd content after regeneration process is similar to the initial sample. It should be noted that the ICP-MS is substantially more sensitive than ICP-OES. We have now included the related information in the revised manuscript.

Supplementary Fig. 23 The Pd loss percentage detected by ICP-MS and the CH₃COOH production rates during cyclic tests (left side). The Pd content in material after cyclic tests is compared with the initial sample, which is detected by ICP-OES (right side).

6. How about the performance of a physical mixture of Pd-WO₃ and PdO nanoparticles for the conversion of CH₄ into CH₃COOH under light irradiation?

Author Response: We thank the referee for his/her valuable suggestion. According to the suggestion, we have prepared PdO-Pd/WO₃ and PdO/WO₃-Pd/WO₃ composites by physical ball-milling, and further compare their photochemical performance with PdO/Pd-WO₃-2 sample. The content of mixed PdO is controlled to approach that in PdO/Pd-WO₃-2 sample. As shown in the newly added Supplementary Fig. 27, the photochemical performance cannot be improved through simple physical mixing of Pd/WO₃ composite with PdO, further corroborating the importance of Pd/PdO interface to CH₃COOH synthesis. We have added the related data and discussion into the revised manuscript.

Supplementary Fig. 27 The comparison of photochemical CH₄ conversion over Pd/WO₃, the mixture of Pd/WO₃ with PdO and the mixture of Pd/WO₃ with PdO/WO₃. The content of mixed PdO is controlled to approach that in PdO/Pd–WO₃-2 sample.

7. The details of experimental procedures should be supplemented for others to repeat them. There is no experimental procedure for the regeneration of the used PdO/Pd-WO₃ in the current manuscript.

Author Response: We thank the referee for pointing out this issue. For recovering the photochemical performance, the sample was calcinated at 230 °C for 3 h with a heating rate of 1 °C min⁻¹ in air. We have added the corresponding experimental procedure into the revised manuscript.

8. The authors should do comprehensive literature search for the synthesis of PdO/Pd-WO₃ hybrid structures to emphasize the novelty and significance of this work in terms of materials chemistry.

Author Response: We thank the referee for his/her thoughtful suggestion. The Pd/PdO interface is an important structure in many research fields such as catalytic ammonia synthesis (J. Mater. Chem. A 2019, 7, 12627–12634), water splitting (Adv. Mater. 2023, 2208860), methanol fuel cell (Adv. Funct. Mater. 2020, 30, 2000534) and gas sensor (ACS Appl. Mater. Interfaces 2020, 12, 42971–42981). Summarized from the related literatures, the methods for preparing Pd/PdO hybrid structure can be sorted as calcination, electrochemical oxidation/deposition, NaBH₄ reduction and laser reduction. In terms of materials synthesis, the electrochemical oxidation/deposition is usually utilized to produce core-shell structure, and the NaBH₄ and laser reduction may not control the Pd/PdO ratio accurately. Therefore, calcination of Pd-based materials is the

promising method for generating high-quality Pd/PdO interface.

In this work, we prepare Pd/PdO interface with abundant grain boundary and certain PdO content by modulating calcination temperature and heating rate carefully, which has not been reported previously. Moreover, we demonstrate that the structural character of the nanoparticle is also important for photochemical CH₄ conversion, which is referred in the Comment 2 from the *Reviewer #3*.

More importantly, we provide an alternative approach for the substitution of additive CO in tradition through the chemical looping in material, demonstrating the novelty and significance of our work in terms of material chemistry. We have cited the related references in the revised manuscript.

Reviewer #2 (Remarks to the Author):

This article by Xiong and coworkers reported a direct synthesis of CH₃COOH solely from CH₄ via photocatalysis without additional reagents. High selectivity and yield toward CH₃COOH are achieved in a photocatalytic flow reactor, along with the novel MVK oxidation mechanism for forming of CO, are excellent features of this work. The study is well conducted, the SI is complete, and the results clearly presented. I think this work deserves to be considered for publication in Nature Comm. However, some important mechanistic concerns should first be addressed before acceptance of this work.*

Author Response: We really appreciate the referee's positive evaluation for our work, and are grateful to the referee for his/her insightful suggestions to help us further improve the quality of our manuscript.

1. CO gas was not detected in the products in this work, but the CO was confirmed by in situ DRIFTS measurements. Why the CO* could not be further oxidated to CO₂?*

Author Response: We thank the referee for his/her insightful suggestion. According to previous works, the adsorption of CO on PdO surface is extremely strong (J. Chem. Phys. 2010, 133, 084704; Acc. Chem. Res. 2015, 48, 1515-1523), and *CO can be converted before desorption from materials surface. In addition, the absence of additional oxidant (e.g., O₂) also inhibits the oxidation of *CO to CO₂. Our experiments show that introducing O₂ into the reaction system can promote the CO₂ production (Supplementary Fig. 15). Therefore, in the absence of additional oxidant, the *CO tends to undergo coupling with the active *CH₃ rather than overoxidation. Similar result has been reported in the recent work of CH₄ oxidation to CH₃COOH (Nat. Catal. 2022, 5, 45-54). We have added the corresponding reference into the revised manuscript.

2. P9, line153-155, 'with the further increase of PdO content, the Pd-WO₃ interface is gradually diminished, which reduces photo-induced charge separation efficiency and substantially suppresses photocatalytic performance', the authors thought that the photo-induced charge separation efficiency decreased with the increase of PdO content. However, this point should be proved by experimental data, such as photocurrent response data.

Author Response: We thank the referee for his/her valuable suggestion. In the revision, we have performed the photocurrents measurements and time-resolved PL (TRPL) decay spectroscopy for the as-prepared samples. As revealed in the newly added Supplementary Fig. 9, the photocurrents are reduced with the increase of PdO content, indicating that the excessive PdO is detrimental to charge separation. Furthermore, the results of TRPL spectra show that the excessive PdO hinders the Schottky contact

between Pd and WO_3 and reduces the electron transfer efficiency (the newly added Supplementary Fig. 10). The related results are also referred in the Comment 4 from the *Reviewer #1*. We have added the corresponding data and discussion into the revised manuscript.

Supplementary Fig. 9 Photocurrent responses of the as-prepared samples.

3. In Supplementary Fig. 21, the most of CH_4 could not be oxidated to CO^* and then to form CH_3COOH in the in gas–solid phase, but transformed to C_2H_6 . Does it mean that more H_2O is helpful for forming of CO^* ? Why?

Author Response: We thank the referee for his/her thoughtful comment. We have performed the gas–solid phase experiments to demonstrate that the flux of H_2O is more favorable for CO^* and CH_3COOH formation than the non-flowing H_2O . As shown in the control experiments (the revised Supplementary Fig. 32), the reactants are CH_4 and water vapor, in which the generated $\cdot\text{CH}_3$ is more able to undergo coupling in gas phase. Improving H_2O content promotes oxygenate production, but CH_4 coupling can still be obviously observed. For this reason, we employ the photochemical flow reaction device to introduce flowing H_2O for controllable methyl–carbonyl coupling and enhance CH_3COOH production.

Supplementary Fig. 32 Photochemical CH₄ conversion performance over PdO/Pd–WO₃-2 in gas–solid phase with different H₂O usage.

4. In this work, the OH radical derived from water oxidation by photo-induced hole, how about the role of the photo-induced electron?

Author Response: We thank the referee for his/her thoughtful suggestion. As revealed in the newly added Supplementary Fig. 20, the conductive band positions of WO₃-based materials are lower than H₂O/H₂ potential. Therefore, the photo-induced electrons can reduce the WO₃ by consuming its lattice oxygen. However, the ratio of consumed oxygen atoms to the total lattice oxygen atoms of WO₃ is determined to be 1.28% through the calculation based on the demand of OH radical production, which is negligible as compare with the consumption of PdO (taking PdO/Pd–WO₃-2 as an example). The lattice oxygen of WO₃ can also be recovered in the regeneration process. We have now included the related data and discussion in the revised manuscript. The related results are also referred in the Comment 3 from the *Reviewer #1*. We have now added the corresponding discussion into the revised manuscript.

Reviewer #3 (Remarks to the Author):

The manuscript “Photocatalytic flow synthesis of acetic acid solely from methane with high selectivity by heterointerface catalyst” demonstrates the synthesis of acetic acid directly from methane using Pd-PdO supported over WO₃. It shows high selectivity to acetic acid, especially in flow conditions, however, the productivity is limited due to the looping process of methane oxidation by oxygen of Pd. The research is well performed and the manuscript clearly represents the discovery. I would recommend it for publication after clarification of several points:

Author Response: We really appreciate the referee’s positive evaluation for our work, and are grateful to the referee for his/her insightful suggestions to help us further improve the quality of our manuscript.

1. First of all, I would not call it “catalysis” because the catalyst cannot be consumed during the reaction as in this case. Thus, I would use the term “photochemical looping”.

Author Response: We thank the referee for his/her insightful suggestion. In the revision, we have revised the title and corresponding description in the manuscript.

2. The effect of the calcination temperature on the Pd/PdO ratio and acetic acid yield looks reasonable, however, it seems strange to me that, for example, samples 2 and 4 containing comparable amounts of O according to TPR have so high differences in the production rate of acetic acid. In fact, PdO is a very flexible material and usually can be easily oxidized and reduced. I would expect an increase in the size of metal nanoparticles or interaction of Pd with WO₃ at high calcination temperatures could be also important. I would recommend to perform the reduction of the sample calcined at high temperature (for example 5) and oxidation at low temperature (200 °C) to check if it brings it to the performance of sample 2. If not the changes are not reversible and have structural character and it would be important to identify them.

Author Response: We thank the referee for his/her thoughtful suggestion. The lattice oxygen in PdO affects both charge separation and intermediate generation. Increasing the PdO ratio from 39.2% to 70.6% (samples 2–4, calculated by the results of H₂-TPR) suppresses the charge separation efficiency (the newly added Supplementary Fig. 10) and inhibits the role in intermediate formation. Therefore, the samples 2–4 show different performance in CH₃COOH production.

Our experiments show that the re-oxidation of the reduced PdO/Pd–WO₃-5 sample can roughly achieve the performance of fresh PdO/Pd–WO₃-2, implying that the construction of appropriate Pd/PdO interface is favorable for CH₃COOH production (the newly added Supplementary Fig. 28).

We further thank the referee for his/her thoughtful comment about the size effect. According to the suggestion, we have compared the photochemical properties of the samples with different particle sizes (the newly added Supplementary Fig. 29). With the same annealing treatments on PdO/Pd-WO₃-2, the lower photochemical properties are observed in both the nanocomposites with particle size of 2 nm and 27 nm, indicating that the main structural character is the optimized Pd/PdO interface. We have added the related data and discussion into the revised manuscript.

Supplementary Fig. 28 The comparison of photochemical CH₄ conversion performance over the reoxidized sample and PdO/Pd-WO₃-2.

Supplementary Fig. 29 TEM images of PdO/Pd-WO₃-2 samples synthesized by (a) photo-deposition and (b) ascorbic acid (AA) reduction. (c) The photochemical CH₄ conversion performance of the as-prepared PdO/Pd-WO₃-2 samples.

3. The authors do not talk a lot about the role of WO₃ in the reaction. What would happen if WO₃ is substituted by TiO₂? Is it important for the stabilization of Pd-PdO interface? It could be important to clarify the effect of semiconductor and perform its characterization (band gap...).

Author Response: We thank the referee for his/her insightful suggestion. According to

the suggestion, we have synthesized the corresponding samples of PdO/Pd–TiO₂-1 to 5 (TiO₂ is purchased from Aladdin, anatase, 30 nm) and conducted the photochemical CH₄ conversion process. As shown in the Fig. R1, the substitution of TiO₂ boosts the production of C₂H₆ and CO₂. However, the CH₃COOH production and selectivity are reduced significantly. The promotion of C₂H₆ and CO₂ production agrees with the previous report of photocatalytic CH₄ oxidation in H₂O (Catal. Sci. Technol. 2017, 7, 635–640).

Notably, the properties of PdO/Pd–TiO₂-5 are approximately comparable with those Pd/TiO₂ and PdO/Pd–TiO₂-1 to 4 samples, which exhibits significant discrepancy with the case of WO₃ supports. An interesting phenomenon is observed that the color of PdO/Pd–TiO₂-5 (yellow) turns grey gradually during the photochemical process, even in Ar condition (Fig. R2a), indicating that the Pd is reduced by the photo-induced electrons of TiO₂ spontaneously. As such, the stabilization of Pd/PdO interface under unreacted condition (i.e., without CH₄ existence) is important so that the interface can be utilized for CH₃COOH synthesis completely. Considering the above reasons, TiO₂ is not the suitable support in this work.

The band structures of the prepared WO₃-based samples are shown in the newly added Supplementary Fig. 20. The conductive band positions of the samples are much lower than normal TiO₂ (-0.3 eV), which can reserve PdO largely (Fig. R2b).

Fig. R1 The comparison of photochemical CH₄ conversion performance over the prepared samples using TiO₂ as support (only for review).

Fig. R2 The photographs of the reaction mixtures containing (a) PdO/Pd–TiO₂-5 and (b) PdO/Pd–WO₃-5 before and after 2 h photochemical process under Ar condition (only for review).

4. *Is there hydrogen generation during the reaction? It could be formed during methane and water splitting.*

Author Response: We thank the referee for his/her thoughtful suggestion. As revealed in the newly added Supplementary Fig. 20, the conductive band positions of the prepared samples are lower than H₂O/H₂ position, indicating that H₂ cannot be generated during the photochemical process. The H atom dissociated from CH₄ will be captured by OH radical to form H₂O. The related discussion is also referred in Comment 3 from the *Reviewer #1* and Comment 4 from the *Reviewer #2*.

5. *There is a peak of CO over Pd at 2060 cm⁻¹ which is usually assigned to CO adsorption over edges. It could be interesting to perform FTIR CO over all samples to check the effect of calcination temperature on the state and morphology of surface Pd atoms.*

Author Response: We thank the referee for his/her insightful suggestion. According to the suggestion, we have performed the CO adsorption DRIFTS measurements over all samples. As shown in the newly added Supplementary Fig. 24, the peaks at 2097 and 1983 cm⁻¹ are ascribed to the linear- and bridge-type CO adsorption on Pd site, respectively. The new peaks arising at 2132 cm⁻¹ over PdO/Pd–WO₃-1 to PdO/Pd–WO₃-3 sample are attributed to the linear CO adsorption on partially oxidized Pd (ACS Catal. 2021, 11, 5894-5905), demonstrating the well-formed Pd/PdO heterointerface in the samples. However, according to the report by Zorn et al., the CO adsorption on fully oxidized Pd is weak (J. Phys. Chem. C 2011, 115, 1103-1111). This leads to the gradual disappearance of the peak at 2132 cm⁻¹ over PdO/Pd–WO₃-4 and PdO/Pd–WO₃-5 samples. Of note, the broad peaks at 2045 cm⁻¹ are observed, which can be assigned to bridge-type CO adsorption over PdO edges with WO₃ supports, corresponding to the detection results of carbonyl intermediate by in situ DRIFTS. We

have added the related data and discussion into the revised manuscript.

Supplementary Fig. 24 CO adsorption DRIFTS spectra of the as-prepared samples.

6. The opposite trends of methanol and acetic acid production could be explained by the intermediate synthesis of methylformate which can be isomerized to acetic acid over Pd catalyst (<https://doi.org/10.1002/cssc.202201006>). It would explain why the selectivity is high in flow conditions at deeper conversion. The migration of CH_3 radical in the flow seems strange to me because the lifetime of radical should be low. It could be interesting to try PdO-Pd/WO₃ for the oxidation of methanol – probably it could be converted to acetic acid.

Author Response: We thank the referee for his/her valuable suggestion. The migration of $\cdot\text{CH}_3$ radical is referenced from the recent publication which states “that chemisorbed CO^* dominates, then react with the flux of CH_3^* being formed by methane oxidation (or the reverse of CH_3OO^* formation (reaction (2))) to form acyl, acetoxy and acylperoxy and hence yield the observed C₂ oxygenate products”. We cite this description to emphasize the cascade reaction between $^*\text{CO}$ and $^*\text{CH}_3$ on catalyst in continuous reaction channels. We have revised the corresponding description in the revised manuscript.

According to the suggestion, we have performed the methanol oxidation experiment under Ar condition using PdO/Pd-WO₃-2 sample. As shown in the newly added Supplementary Fig. 34, increasing methanol concentration is beneficial to produce CH₃COOH. However, such a high concentration of methanol is not existing in the routine test in our work. It should also be noted that our flow synthesis of CH₃COOH does not involve iodide for stabilizing carbocation, which is a requirement for methylformate isomerization as mentioned in the reference

(<https://doi.org/10.1002/cssc.202201006>). Thus, we believe that the generation of CH_3COOH mostly originates from CH_4 conversion. We have added the related data and discussion into the revised manuscript.

Supplementary Fig. 34 The comparison of photochemical methanol oxidation performance over PdO/Pd- WO_3 -2 with different methanol concentrations under Ar condition.

7. Recently the photocatalytic synthesis of acetic acid by carbonylation of methane has been published in *J. Am. Chem. Soc.* 2023, 145, 2, 1185–1193. It would be important to refer to this work.

Author Response: We thank the referee for his/her valuable suggestion. We have now cited this work (Ref. 15) in the revised manuscript.

REVIEWERS' COMMENTS

Reviewer #1 (Remarks to the Author):

The authors have properly addressed all the issues, which this reviewer raised, in the revised manuscript.

Reviewer #2 (Remarks to the Author):

The revised manuscript has been improved and it is suggested to be published.

Reviewer #3 (Remarks to the Author):

The authors have responded correctly to all my comments on this manuscript. I recommend its publication in its present form

Point-by-point response to the reviewers' comments

Reviewer #1 (Remarks to the Author):

The authors have properly addressed all the issues, which this reviewer raised, in the revised manuscript.

Author Response: We really appreciate the referee's positive evaluation for our work, and are grateful to the referee for his/her insightful suggestions to help us substantially improve the quality of our manuscript.

Reviewer #2 (Remarks to the Author):

The revised manuscript has been improved and it is suggested to be published.

Author Response: We really appreciate the referee's positive evaluation for our work, and are grateful to the referee for his/her insightful suggestions to help us substantially improve the quality of our manuscript.

Reviewer #3 (Remarks to the Author):

The authors have responded correctly to all my comments on this manuscript. I recommend its publication in its present form.

Author Response: We really appreciate the referee's positive evaluation for our work, and are grateful to the referee for his/her insightful suggestions to help us substantially improve the quality of our manuscript.